# *Divinas tetas*: Doing Theology from Mutilated Bodies

André Sidnei Musskopf [1,*] and Ana Ester Pádua Freire [2,*]

1   Department of Religious Studies, Universidade Federal de Juiz de Fora, Juiz de Fora 36036-900, Brazil
2   Independent Researcher, Belo Horizonte 31630-900, Brazil
*   Correspondence: asmusskopf@hotmail.com (A.S.M.); anaesterbh@gmail.com (A.E.P.F.)

**Abstract:** The article is an exercise in constructing theology from a Latin American perspective in dialogue with queer studies and theologies. The starting point is the tragic death of Lorena Muniz, a transwoman, in the process of getting breast implants making evident aspects of gender oppression, cis-sexist aesthetic pressure, and state neglect of health care specific to the trans population as denounced by ANTRA (National Association of Travestis and Transexuals). From this context, the article discusses a hermeneutics of mutilation in relation to Latin America and to the experience of trans people and introduces countersexuality (Preciado) as a way to resist the mutilations of cis-heteropatriarchy. With those tools in hand, the last part of the article realizes an exercise of theological and religious imagination engaging with the song *Vaca profana* (Caetano Veloso/Gal Costa) as a possible way of reconciling the reality of Lorena Muniz' death with the hope for a different future through the symbolism of the *divinas tetas* [divine tits].

**Keywords:** *Vaca profana*; hermeneutics of mutilation; countersexuality; queer theology

In memory of Lorena Muniz and Gal Costa, *vacas de divinas tetas.*

## 1. Introduction

Marcella Althaus-Reid (2000) has stated that doing Liberation Theology is a "risky affair"—and it is more so when it dares to question the sexual assumptions of such a theology (Córdova Quero 2010). In this case, the stakes are even higher since we dare to engage with a reality which we do not experience first-hand or are contemplating ourselves—the implant of breast prosthesis in the process of gender assignment. It could be argued that we are reflecting on an experience that is not our own and over which we have no authority—and we would agree.

In our defense—would we ever need to defend ourselves to someone who actually has such authority—we should state that we do not imply that it is our own lived experience, nor do we claim to be speaking about that experience itself. We are, though—and we commit to that—speaking of how such a reality impacts our own experiences and how we feel affected by it in our embodiment. If that is possible, we feel connected to such experience through the common norms and regulations governing the lives and bodies of us all, particularly those identified as sexual and gender dissidents.

In this perspective, what we seek to accomplish in this article is an exercise of self-reflection in which we question gender and sexual norms from a theological and religious point of view. The cruel reality experienced by trans people in Brazil prompts us to question the systems that produce such violence and seek ways to imagine other possible worlds. Although this exercise is realized in the context of a symbolic order, the necessary acts of change in the material life are not ignored. Akin to Mary Hunt's reflection (Hunt 2000) on sexual ethics, this exercise in reconciling the life and death of Lorena Muniz through the reading (and singing) of the song *Vaca Profana*, should act as "renewable moral energy" (Hunt 2000)[1] and fuel our "work for love" (Harrison 1985).

## 2. From *tetas* to Ashes—Gender Oppression, Cissexist Aesthetic Pressure, and State Neglect of Health Care

On 21 February 2021, the *Associação Nacional de Travestis e Transexuais*, ANTRA, (National Association of *Travestis*[2] and Transsexuals) published a "public note in mourning for Lorena Muniz and on trans health" (ANTRA 2021), that stated:

Lorena left Pernambuco for São Paulo in order to realize her desire to have surgery to have breast implants (prostheses). However, on the day of her surgery, videos posted on the internet announced that the clinic had faced an accident involving fire and during the evacuation of the building, Lorena was left behind by those responsible for her surgery.

According to information from people present at the event, she was still sedated awaiting surgery and when she was abandoned by the clinic staff, she ended up inhaling a high quantity of smoke and carbon dioxide gas, which caused a worsening in her health. She was rescued and transferred to a hospital in a very serious condition and unfortunately, she was unable to recover and she died. We also remember that in the first news broadcast by the media about the fire at the clinic, it was reported that there were no victims and there were also several attempts to make invisible, to hinder or hide the serious situation that affected Lorena. Also noteworthy was the clinic's negligence in not providing any support to the victim or her family.[3]

Lorena Muniz is just one more victim of a vicious cycle that leads to the early death[4] of trans people in Brazil, the country with the highest rates of violence and murder of trans people in the world[5]. She was not murdered on the streets, in a dark alley, in her room or in broad daylight—all too common locations where trans people lose their lives. She was left to die in a healthcare facility consumed by fire, erased from the news and denied any form of recognition of her existence. Her life was not a livable life; her death was not a grievable one (Butler 2009). Her desire "to be more authentic as [she] looked more like what [she] dreamed of herself"[6] ended up in the ashes of a burned down-building.

ANTRA's note brings her back to life as a preferred subject for our theological doing. Not only by noticing her death, but also by pointing out at least three interrelated issues that led to it. According to the note: "Lorena is yet another victim of **gender oppression**, **cissexist aesthetic pressure**, and **state neglect of health care** specific to the trans population" (ANTRA 2021 [bolds added]). These three factors are part of the axes of oppression that operate in the daily reality of trans people in Brazil, together with class and race.

**Gender oppression** presents itself as a regulation of bodies based on the binary division (man/woman) that envisions eliminating everything that escapes or exceeds the cisheteronormative standard. Patriarchy is not only sustained by the image of the white, cisgender, heterosexual man, but also by the image of an "other" of this representation—the essentialized and universalized woman. After all, patriarchy needs this binarism to create and maintain inequalities and hierarchies, and trans people make explicit the artificiality of the construction of such binarism (Tatton-Schiff 2022, p. 295).[7]

According to Rita Laura Segato (2012):

In Modernity's world, there is no duality, there is dualism. While as in duality the relation is of complementarity, the binary relation is supplementary; one term supplements the other and does not complement it. When one of these terms becomes "universal", that is, of general representativity, what was hierarchy is transformed into the abyss, and the second term is converted into rest and residue: this is the binary structure, different from the dual one. According to the colonial modern and binary pattern, any element, to reach the ontological plenitude, plenitude of being, must be equalized, that is, equated from a grid of common reference or universal equivalent. (pp. 46–47)

Patriarchy acts under the logic of universalization and essentialism, prescribing gender from static models of representation of man and woman. In this sense, gender oppression

also takes place through the denial of the plurality and of the recognition of the diversity of gendered experiences.

In this hierarchization of life, a **cissexist aesthetic** is instituted based on the same binary standards of man and woman. Everything that diverges from this aesthetic is considered a pathology, a crime and/or a sin. For many trans people, bodily changes are part of the process of becoming who they understand themselves to be. This may include having breast implants or their removal and other body modifications. According to Tatton-Schiff (2022):

> When an individual whose sense of gender identity does not fit within the cultural rules imposed upon them is offered no support in their 'otherness', it should be no surprise that they may come to relate more strongly to the supposed 'opposite' gender identity, consequently perpetuating the binary that has so oppressed and failed them. Some may also experience a strong attraction towards how they believe those with the 'opposite' gender delineation experience their gender identity and role. [T]hese anxieties and desires may then feed a self-perpetuating, patriarchal cycle of castigation and control. (p. 301)

Whatever the reasons for someone to decide on body modifications in relation to gender norms, in a cis-heteropatriarchal society—and health care system—these procedures, in general, include high risks and are not done in a safe way when accomplished by trans people. That's one of the reasons why the trans population has **specific health care demands**. Although those specific needs go far beyond processes of gender reassignment[8] (surgeries of sex and gender affirmation), those are an important part for many trans people and may include:

> [m]astectomy (removal of the breasts), breast plastic surgery (implantation of silicone breast prosthesis), thyroplasty (reduction of the Adam's apple to feminize the voice), hysterectomy (removal of the uterus and ovaries), the surgeries of sexual reassignment in the genital organ of both genders, and complementary surgeries. (Silva 2021)[9]

Specifically in the case of *travestis*, many have resorted to using industrial silicone to reach the "perfect body."[10] The documentary *Inacreditável: casa da Bartô* [Unbelievable: House of Bartô] by Andrade and Maio (1987), presents the *travesti* Bartô—a famous *Bombadeira*[11]—who operated a clandestine business in the city of São Paulo, Brazil, which injected its clientele with industrial silicone. "House of Bartô" worked as a clandestine clinic for this type of procedure, which, in addition to being contraindicated, took place in terrible unsanitary conditions. The documentary narrates the experiences of many *travestis* with industrial silicone for breast, buttock and face implants and the effects of these implants on their bodies. Being highly toxic, industrial silicone causes swelling, nodule formation, pain, and inflammation. Many of the *travestis* who participated in the filming had deformed bodies, and removing the silicone was almost impossible because it adheres to other organs. This kind of procedure is still very common among *travestis*, especially the poorest ones (Pinto et al. 2017).

In this context, staying alive and daring to dream of feeling comfortable in one's own body is an act of resistance and a revolutionary act of love that can serve our theological imagination for thinking of other possible worlds. Theology—and queer theology, for that matter—has to be able to reconcile Lorena's desire for having *tetas*[12] in a world of justice and peace for everyone with the reality of fire and ashes that reminds us of the *cystemic*[13] oppression and violence that mutilates and fragments our bodies to exercise control. Restoring and acknowledging the sacred place of *tetas* in the life of a transwoman might help us to look at our gender and sexual misery (Cardoso 2001), opening up new spaces to experience and talk about the divine[14].

### 3. *Tetas* and Mutilations: Latin American and Trans Perspectives

The body is central to any form of theology, and this issue has been abundantly raised by different forms of "body theology."[15] It is not an innocent or minor issue, as some more universalist theologians would argue. The body is the primary location for any theological reflection, not only as the site in which humans experience the world in all its dimensions and in relation to oneself, to others, to the Earth, and environment and to the divine[16], but also because, mostly, some/bodies are assumed as the "norm" and used to measure and control other/bodies through pretentious universalist forms of theology (male, white, Northern, cisgender, heterosexual). According to Peter Nash (2003):

> It is a pretentious illusion that there is something pure and objective about the way theology has been done in Western church, as if it were handed down directly by the Almighty to the theologians of the correct methodology. Somehow, the great fathers of theology from the first century are presented as having practiced their craft without any social context, and then suddenly, Asians, women, Blacks, gays, and Latin Americans began infecting the theological purity with their bodies and their body-oriented questions and assertions. (pp. 25–26)

In a very literal way, Marcella Althaus-Reid (2000) describes how the colonial project imposed in Latin America operated in relation to the body and the theology resulting from it. At the beginning of her project of "indecenting theology," she states:

> From some people the buttocks were cut, to others the thighs, or the arms . . . cutting hands, noses, tongues and other pieces from the body, eaten alive by animals and (cutting) women's breasts' (Todorov 1987, p. 151). These mutilation rituals, paraphrasing Lacan, could be compared to the cutting off of the breasts of truth, the reductionism into a new bodily order, that is, humanity reduced to one formula, one law of union and compulsion. This required a massive mutilation. The need for Grand Narratives always takes with it some cuttings and mutilations in itself. Latin American theology comes from that, a mutilation of symbolic knowledge such as theology, politics, economics, science and sexuality. (Althaus-Reid 2000, p. 11)

This way, any Latin American—or anti-colonial—reflection will necessarily have to deal with the heritage of mutilation as constitutive of coloniality—still very much alive. A history of violence is evident in the fragmentation of bodies. Marcella argues in a reflection about cosmetic surgery (Althaus-Reid 2008) that such mutilations have been relieved incessantly in the lives and bodies of women.[17] Therefore, she proposes a hermeneutics of mutilation that can identify the pedagogical effects of such mutilations and find ways to reconcile the goodness of all bodies. According to her:

> Mutilative hermeneutics confer deep meaning on acts of the cutting of the genitalia, the specific covering of the body to disguise/modify the natural female appearance, the seclusions and rituals of extreme cleansing during women's bleeding experiences such as menstruation and after birth. All have left their marks in the body of the believer. These are not just scars, but hermeneutically speaking, they are pedagogical procedures. (Althaus-Reid 2008, p. 72)[18]

Judith Tatton-Schiff (2022) reflects on body modifications in the experience of trans people from the perspective of Marcella's "hermeneutics of mutilation". They are aware that "gender transition surgery would rarely be deemed 'cosmetic' by those who undergo it, rather being experienced as 'corrective' or even 'life-saving" (Tatton-Schiff 2022, p. 304). Instead, they see it as "a caution to those with a trans+ identity or gender dysphoria 'buying into' heteronormative, patriarchal 'goods', however unknowingly or unintentionally, as they envisage their own individual 'ideal selves" (Tatton-Schiff 2022, p. 304), as in the cissexist aesthetics discussed above.

Tatton-Schiff's concern is not with the moralizing of body modifications (of trans people or anyone else), but with the "body of heterosexual systems' identified by Althaus

Reid; the polarized, oppressive, binary vision of 'what is a man' and 'what is a woman"
(Tatton-Schiff 2022, p. 304). Drawing from Liberation, Disabled, Body, Feminist, and
Womanist Theologies, Tatton-Schiff discusses the body politics that ground the theological
doing when it identifies the "perfect body" pushing all other forms into a constant struggle
to get as close as possible to that ideal. Thus, they conclude:

> Regardless of the decisions that individuals might make around their own bodies
> and their modification, this surely must be the definitive Liberation/Christian
> approach to trans-thinking; claiming and celebrating 'nonconventional bodies'
> and gender expressions within a truly holistic, Body of Christ approach to human
> experience. (Tatton-Schiff 2022, p. 300)[19]

Thus, it is not that bodies are different and undergo changes that are in question, but
the oppressive systems that forcefully and deliberately or unconsciously and surreptitiously
impose the need for changes for "nonconventional bodies" to be accommodated or tolerated.
From a decolonial perspective, that is to say, the articulation of capitalism, racism and
patriarchy (Segato 2021). Therefore, as Althaus-Reid (2017): "In theology, the disintegrated
appears homogenized, even though the critical reality of our lives shows precisely the
contrary: where we most coherence assume, more fragmentations we hide" (p. 130).[20]

We cannot know what exactly Lorena Muniz was seeking or what were her deepest
wishes and motivations when she left Pernambuco to travel to São Paulo[21]. All we know
is that she wanted to get breast implants, for which she longed and had prepared for a
long time. She wanted to change her body to feel more like what she dreamed of herself
as an act of transformation since, as Althaus-Reid (2008) states, "To give hospitality to our
own fragmentations may require sometimes acts of transformations" (p. 114). As we have
seen, her desire for *tetas* has led her to "ashes"[22]—as so much of the reality of life in Latin
America[23]. But rising from the dead, she may teach us something about how to deal with
the multiple mutilations and counteract the oppressive systems that led to her death and
envision a theology for *every*-body.

#### 4. *Tetas* and Dildos: Mutilating the Mutilation

A common debate in the preparation of Pride Parades or in comments afterward is the
scandal of *travestis* showing their *tetas* during such events. Most of the concerns expressed
are related to the idea that this act might scandalize the viewers, particularly "families and
children", as they explicitly and proudly rub the built-in prosthetics in society's face. This
concern contrasts with Brazil's carnival festivities in which it is not uncommon to have
topless women in Samba School Parades[24].

Helena Vieira (2020), a transactivist in one of her classes, mentioned a situation in
which *tranvestigênere*[25] Indianare Siqueira, who had been identified in the past as a *travesti*,
uncovered her *tetas* during a demonstration. Because her *tetas* were uncovered, Indianare
was taken to the police station accused of indecent exposure. At the police station, she
claimed she was doing nothing wrong and challenged the policeman by using her identity
card. On Indianare's ID card, she was listed as a man. Indianare then asked, what would
be the problem for a man to be shirtless? Some *tetas* crash the cystem.

Paul Preciado's concept of a dildo might help to think about the power of the *tetas* as a
countersexual act. For the philosopher,

> The dildo is not an object that replaces something that's missing. It is a cutting-
> and-pasting operation that takes place within heterosexuality, displacing the
> supposed organic center of sexual production onto a space outside the body. The
> dildo, as a reference of power and sexual arousal, betrays the anatomical organ
> by moving into other signifying spaces (organic and inorganic, male and female)
> that are resexualized by dint of their semantic proximity. From that moment on,
> anything can become a dildo. All is dildo. Even the penis. (Preciado 2018, p. 66)

To state that *tetas* are also dildos implies acknowledging how their prosthetic character
operates in the gender prescriptions. After all, the fixity of gender affirms that someone

with *tetas* is a woman and someone without them is a man. However, in the face of the mutilation of bodies—compulsory or not—this cutting-and-pasting operation also takes place in the *tetas*, revealing how they are a biopolitical technology. For Preciado (2018), a biopolitical technology is "an element within a complex system of regulating devices that define relationships between bodies, tools, signs, machines, uses, and users" (pp. 64–65).

Just as "the dildo can be considered a critical act in the history of countersexual technology" (Preciado 2018, p. 65), the *tetas* can also be. Breast augmentation, breast reduction, and breast reconstruction are examples of the manipulation of the body through technologies of gender. It is something that is added, or subtracted, that acts in the construction of gender. But, instead of thinking that *tetas* strengthen gender stereotypes, it is possible to state that they—like the dildos—have a role in countersexuality, since, by being "cut-and-pasted," they re/configure the bodies in the cis-heteronormative regime. On countersexuality, Preciado (2018) explains:

> The name "countersexuality" comes indirectly from Michel Foucault, for whom the most efficient form of resistance to the disciplinary production of sexuality in our liberal societies is not the fight against prohibition (as the antirepressive sexual-liberation movements of the 1960s proposed), but rather counterproductivity—that is to say, the production of counter- protocols and forms of pleasure—knowledge as alternatives to the disciplines of the modern sexual regime. The countersexual practices proposed here should be understood as technologies of resistance or, put another way, as forms of sexual counterdiscipline. (p. 21)

Is a trans body that implants 500 mL of silicone to create a pair of *tetas* adapting to what is understood to be the pattern of femininity, or is it challenging femininity? Perhaps the answer is neither one nor the other, since the silicon implant, when understood as a "technology of resistance", operates outside the logic of femininity by creating other bodies. The challenge of the analysis that a hermeneutics of mutilation proposes is to notice how bodies are re/created outside the system in which identities are acknowledged. In this sense, when Preciado (2018) states that "An organ's name *always* has prescriptive value" (p. 114), we state that *not always*. For example, on the body of a cisgender women, a decent understanding can see the *tetas* producing milk (emphasizing the productive and reproductive role of women's bodies in patriarchy and capitalism), while on the body of a trans-woman the *tetas* can be seen purely as an (indecent) means for pleasure (through the hypersexualization of trans women's bodies).

> That is why gender is prosthetic!

> That is, it does not occur except in the *materiality* of the body. It is entirely constructed, and, at the same time, it is purely organic. It springs from the Western metaphysical dichotomies between body and soul, form and matter, nature and culture, while simultaneously tearing them apart. Gender resembles the dildo. Both surpass imitation. Their carnal plasticity destabilizes the distinction between the imitated and the imitator, between the truth and the representation of the truth, between the reference and the referent, between nature and artifice, between sexual organs and sexual practices. (Preciado 2018, p. 28)

Gender is not only prosthetic, but it is also mutilation. It is re/configured all the time by sexual bodies that challenge the binary system. *Tetas*, when seen in this il/logic, denounce, challenge, and dislocate the rigid and fixed structure of the bodies, showing that cutting-and-pasting operations are happening all the time. So, it is possible to say that when Lorena Muniz entered the operating room to get her *tetas* it was not only her body that was going to be mutilated by the surgery, but that she was mutilating the cis-heteronormative regime itself. Her body is the most visible expression of the mutilation of a regime that insists on categorizing the world in oppositional binaries. Lorena Muniz intended to cut the codes of gender and, as in a bricolage, wanted to paste on new bodily experiences.

In the face of the mutilations that trespass us all, we want to dare to envision a theology of *divinas tetas*. A theology that, starting from this cutting-and-pasting operation, reflects

on the *tetas* as a possible locus for a theological doing that denounces the "gender parody" (Preciado 2018) and recognizes the *tetas* as divine (outside the cis-heteropatriarchal pattern). And we propose to do that in dialogue with the sacred text of traditional Brazilian Popular Music.

### 5. *Tetas* of Desire: *Vaca de Divinas Tetas*

Theology, as a religious discourse, is an exercise of imagination. A language through which religious experiences are expressed and recreated (Radford Ruether 1993). As in any other human experience, the materials for this exercise (and the experience itself) are found in daily life and acquire a symbolic power through which one feels connected (*religare*) to the divine (that which is beyond the immediate).[26] As language, hegemonic theologies, and religion ab/use this symbolic power in acts of violence that perpetuate oppressive systems. However, as language, theological imagination also implies that "the same words that intend to hurt can equally miss their target and produce an effect contrary to the desired one" (Butler 2009, p. 148).[27]

In the exercise that we propose in connection to the life and death of Lorena Muniz, we bring together many of those materials from daily life. As we have been doing all along, narratives, characters from movies, literature and political activism, songs, poems, news reports, scholars from different fields, and our own personal trajectories are our sources for theological and religious imagination. This affirms our own set of traditions and sacred and liturgical texts, our own "toolbox" (Segato 2021), and our authority as sexual dissidents (Koch 2001). From the reality of mutilated bodies and the restoring power of the *divinas tetas* we start this exercise with the song *Vaca profana* [profane cow].

*Vaca profana* was composed by Caetano Veloso for his friend Gal Costa in 1984. According to Pereira Torres (2016):

> Recorded by Caetano Veloso in 1986 in acoustic version, "Vaca profana" had been composed in 1984, the year in which it was released by Gal Costa in the LP *Profana*, by the label RCA. When released by Gal, the song had its public execution prohibited by the D.C.D.P. (*Division of Censorship of Public Entertainment*). The argument was that the lyrics of the composition attacked the moral and good manners in verses such as: "Spill the good milk on my face". (p. 86)

The relationship between dictatorships and the control of bodies through sexual and gender norms—"moral and good manners"—is very well known.[28] Yet, the attention to the verse used to justify the censorship of the song makes evident just how familiar censors are with a free sexuality and how such a free sexuality terrorizes oppressive systems (religious, political, and economic).

The song's lyrics present a bricolage of references of "the reflections of an enunciator in face of the aesthetic fruition in a universe of imagetic constructions characterized by an urban experience" (Pereira Torres 2016, p. 89). In this perspective, "the first verse is set up as an introduction and the four subsequent verses develop the idea of a poetic subject in face of religiosity, art, pop culture and a world in process of globalization" (Pereira Torres 2016, p. 89). The poetic subject, Caetano himself, writes from his experience of forced exile and imagines/creates a world where there is milk—and honey—for all.

Religion, sexuality, and art are expressed through symbolic systems that rub against each other as expressions of human experience with the divine (Croatto 2001). Although Pereira Torres (2016, p. 92) argues that there is a "desacralisation" in the way the composer articulates the sacred and the profane,[29] from a Latin American Liberation Theology perspective, we claim the "irreverent fidelity to the gospel, [which] deconstructs the systematic and exegetic compartments and finds the sacred irreversibly in the world, and in the world of the poor" (Cardoso et al. 2006, p. 6).[30] With Caetano we "bring, once again, the sacred to the sphere of the profane, the divine to the earthly" (Pereira Torres 2016, p. 97), and theology is relevant again. Desacralization is good—as milk spilled in the face and throat of the "uptights" [caretas].

In this process, not only the separation of sacred and profane but any binarism that supports patriarchy and other derived systems of power (Segato 2012) is overcome by disjunction[31]:

> Through the articulation of antagonic pairs, sacred/profane, tear/laugh, good/bad, close/far, shy/fussy, all the milk/no milk at all, the perspective of disjunction between the subject and a world that does not enable completeness is strengthened (...). Such a disjunction however, does not entail unhappiness: on the contrary, allows him to live the human experience in the multiplicity of its experiences. (Pereira Torres 2016, p. 90)

The lyrics acknowledge the "grace" and "disgrace" in the world and question the oppressiveness of traditional Christian values, all the while witnessing the goodness of happiness. In the song: "The first verses, initiated with a verb in the present tense inflected in the first person, assert themselves as a witness to the commitment to happiness. So, once more, a negative against the values of Christian morality, condensed in the earthly suffering as a form of atonement for the guilt for the original sin, in exchange for a heavenly existence in the life after death is asserted" (Pereira Torres 2016, p. 91).[32]

Although not much is said about the *tetas* themselves, descriptively or conceptually—except for "astonishing" in the last verse –, the whole song is articulated around the subject who holds them and what flows from them. "Alternating the initial terms *cow, owner*, and *goddess*" (Pereira Torres 2016, p. 88), the song points to the sacred character of the profane singer—Gal Costa herself[33]—and her potential in bringing forth change. Before wandering all over the world, bringing together religions and cultures, the composer calls upon this deity to "put your horns out and above the herd".

> In this verse, the critical reference to the erasure of the identity differences is evident in an explicit way: the author uses the metaphor of herd [manada], already crystallized as a form of definition of mass culture that equalizes and reduces the possibilities of human expression. It is important to notice that the collective of cows and oxen would be "flock" [boiada], a term that would adapt perfectly to the metrics, as well as to the lyrics and melody of the composition. Caetano, however, chooses the collective of elephants—"herd" [manada]—which lead us to believe in a double differentiation: the profane cow stands out, not only for putting her horns out, but also for being of an intrinsically diverse nature from those with whom she walks. (Pereira Torres 2016, pp. 92–93)

The song and its history are certainly open to multiple uses and interpretations. In dialogue with the analysis and reflection accomplished by Pereira Torres (2016), we highlighted a few that we believe can help reclaim the countersexuality in Lorena Muniz' desire for having *tetas* in an exercise of doing theology from mutilated bodies. In this cut-and-paste process, we recover and affirm the symbolic power of the *divinas tetas*, hoping blessed by the milk that flows from them. According to Pereira Torres (2016):

> In its totality "Vaca profana" functions as a claim to the woman/cow sacred/profane to express the hopes and desires of the subject, whose words inscribe themselves, therefore, in the singing of the one destined to be his spokesperson. Configured metaphorically as "good milk", the product of the profane cow, that is, the singing of the sacred woman, it is spilled initially only in the face and throat of the poetic subject to, progressively be spilled on everyone who listens to her. In parallel, the "bad milk", initially destined to the "uptights" [caretas], is substituted by the good milk, finally spilled over everybody. (p. 89)[34]

The image of *tetas* spilling milk breaks away from the traditional gender parody for many reasons. It is not like Caetano Veloso is actually expecting Gal Costa to go around gushing milk; however she might be able to "produce" it. Or is he? What "milk" does he have in mind or might be suggested so much as to raise the concerns of censorship? The cow, the *tetas*, and the milk gain meaning in a utopian (religious) experience—radically different from the context (now and then). Perhaps those are the *tetas*—*divinas tetas*—that Lorena Muniz dreamt of and worked for. Not as a definite and out-of-time salvation, but as

a possible redemption in her daily life—an act of transformation: precarious and provisory. By seeking that, she might teach us—and theology—something of the reality of the divine in the midst of the mutilation and fragmentation that marks so much of our experiences (theologically, politically, and economically).

During a *bibliodrama*[35] workshop a few years ago, the participants were asked to mold in clay something they dreamt of. The workshop took place at a Roman Catholic Seminary in a small and conservative region in Southern Brazil. One participant that was very shy and timid during the whole workshop stood out for their concentration on this specific activity. When getting closer to what they were molding, one could see two giant rounded forms slowly and sensually turning into a particular shape: a pair of *tetas*. No one asked her specifically about what they were designing or why they were doing it—although they were praised for its beauty and strength. It was clear that, by designing it, they were not only expressing materially the dream of becoming what they dreamt of themselves, but devotedly creating a material theology for mutilated bodies: *tetas! divinas tetas! derrama o leite bom na nossa cara!* We pray with/to Lorena and Gal Costa.

### 6. Concluding Remarks

While men create altars for their golden calves, we enter the herd of *vacas profanas* [unholy cows]. From there, from the herd's embodied—and transgressive—experience, we produce a theology from mutilated bodies. Mutilation is the denunciation of a colonizing system that mutilated our territory in the past (Althaus-Reid 2000) and continues mutilating today, and it is also the violence of gender oppression, cissexist aesthetic pressure, and state neglect of health care specific to the trans population. However, mutilation can also be countersexuality in action, cutting and pasting other bodies into the theological making.

As queer theologians, we intend to mutilate the cis-heteropatriarchal regime that regulates and is regulated by hegemonic theology. With Lorena Muniz, we implant prostheses that transform the theological body of Tradition. With Gal Costa, we sing death and life in an act of resistance through art—which is fundamental for the queer theology that we propose—one that imagines other forms of sociability.

We are in desperate need of *divinas tetas* [divine tits] to gush milk over the *caretas* [uptights] in face of the extreme right fundamentalist religious neoliberal colonial capitalism that is, once again, mutilating our bodies, the body of the world, and the body of God. And we are also in desperate need of the *divinas tetas* to pour good milk on our faces so that we can continue resisting the forces of death. That is the invitation and the challenge: daring to acknowledge the power, receiving the gift, and acting for justice. Suck it up! *¡Chúpatela!*[36]

### 7. Epilogue

| Vaca profana[37] | Profane cow |
|:---:|:---:|
| Respeito muito minhas lágrimas | I respect very much my tears |
| Mas ainda mais minha risada | But even more my laugh |
| Inscrevo, assim, minhas palavras | I inscribe, this way, my words |
| Na voz de uma mulher sagrada | In the voice of a sacred woman |
| Vaca profana, põe teus cornos | Profane cow, put your horns |
| Pra fora e acima da manada | Out and over the herd |
| Vaca profana, põe teus cornos | Profane cow, put your horns |
| Pra fora e acima da man... | Out and over the her.. |
|  |  |
| Ê, ê, ê, ê, ê, | Ê, ê, ê, ê, ê, |
| Dona das divinas tetas | Owner of divine tits |
| Derrama o leite bom na minha cara | Spill the good milk in my face |
| E o leite mau na cara dos caretas | And the bad milk in the face of the uptights |

| Segue a "movida Madrileña" | Follow the "Madrileña move" |
| Também te mata Barcelona | Also kills you Barcelona |
| Napoli, Pino, Pi, Paus, Punks | Napoli, Pino, Pi, Paus, Punks |
| Picassos movem-se por Londres | Picassos move through London |
| Bahia, onipresentemente | Bahia, omnipresently |
| Rio e belíssimo horizonte | Rio and most beautiful horizon[38] |
| Bahia, onipresentemente | Bahia, omnipresently |
| Rio e belíssimo horiz . . . | Rio and most beautiful horiz . . . |
| | |
| Ê, ê, ê, ê, ê, | Ê, ê, ê, ê, ê, |
| Vaca de divinas tetas | Cow of divine tits |
| La leche buena toda en mi garganta | La leche buena toda en mi garganta |
| La mala leche para los "puretas" | La mala leche para los "puretas" |
| | |
| Quero que pinte um amor Bethânia | I want a love Bethânia to show up |
| Stevie Wonder, andaluz | Stevie Wonder, andaluz |
| Como o que tive em Tel Aviv | Like to one I had in Tel Aviv |
| Perto do mar, longe da cruz | Close to the sea, far from the cross |
| Mas em composição cubista | But in a cubist composition |
| Meu mundo Thelonius Monk's blues | My world Thelonius Monk's blues |
| Mas em composição cubista | But in a cubist composition |
| Meu mundo Thelonius Monk's . . . | My world Thelonius Monk's . . . |
| | |
| Ê, ê, ê, ê, ê, | Ê, ê, ê, ê, ê, |
| Vaca das divinas tetas | Cow of divine tits |
| Teu bom só para o oco, minha falta | Your good only for the hollow, my lacking |
| E o resto inunde as almas dos caretas | The rest flood the souls of the uptights |
| | |
| Sou tímido e espalhafatoso | I am shy and fussy |
| Torre traçada por Gaudi | Tower drawn by Gaudi |
| São Paulo é como o mundo todo | São Paulo is like the world to me |
| No mundo, um grande amor perdi | In the world, a great love I lost |
| Caretas de Paris e New York | Uptights of Paris and New York |
| Sem mágoas, estamos aí | No hard feeling, here we are |
| Caretas de Paris e New York | Uptights of Paris and New York |
| Sem mágoas estamos a . . . | No hard feeling, here we . . . |
| | |
| Ê, ê, ê, ê, ê, | Ê, ê, ê, ê, ê, |
| Dona das divinas tetas | Owner of divine tits |
| Quero teu leite todo em minha alma | I want all the milk in my soul |
| Nada de leite mau para os caretas | No bad milk for the uptights |
| | |
| Mas eu também sei ser careta | But I also know how to be uptight |
| De perto, ninguém é normal | Close up, no one is normal |
| Às vezes, segue em linha reta | Sometimes, following a straight line |
| A vida, que é "meu bem, meu mal" | Life, which is "my good, my evil" |
| No mais, as "ramblas" do planeta | Besides, the "ramblas" of the planet |
| "Orchta de chufa, si us plau" | "Orchta de chufa, si us plau" |
| No mais, as "ramblas" do planeta | Besides, the "ramblas" of the planet |
| "Orchta de chufa, si us . . . | "Orchta de chufa, si us . . . |
| | |
| Ê, ê, ê, ê, ê, | Ê, ê, ê, ê, ê, |
| Deusa de assombrosas tetas | Goddess of astonishing tits |
| Gotas de leite bom na minha cara | Drops of good milk on my face |
| Chuva do mesmo bom sobre os caretas... | Rain of the same one on the uptights . . . |

**Author Contributions:** The authors contributed equally to the manuscript. All authors have read and agreed to the published version of the manuscript.

**Funding:** This research received no external funding.

**Institutional Review Board Statement:** Not applicable.

**Conflicts of Interest:** The authors declare no conflict of interest.

## Notes

1   Hunt takes the expression from (Maguire 2000).

2   *Travesti* (Portuguese) is a gender identity from Latin America that usually refers to a transgender woman. However, the word *travesti* refers to more than gender identity, taking into consideration other intersectionalities, such as race and class. In the past, *travesti* was used to refer to transgender women who were sex workers; today it is more related to a political position. According to Pierce (2020, p. 306) *travesti* "refers most frequently to people assigned male sex at birth and who feminize their bodies, dress, and behavior; prefer feminine pronouns and forms of address; and often make significant bodily transformations by injecting silicone or taking hormonal treatments but do not necessarily seek sex-reassignment surgery".

3   All the quotations in other languages were translated by the authors.

4   According to Alves (2021), the life expectancy of trans people in Brazil is 35 years old.

5   The *Dossiê dos Assassinatos e da violência contra pessoas Trans em 2021* (Dossier of Murders and Violence against Brazilian Trans People in 2021), also produced by ANTRA (2022), portrays the reality that trans people in the country face every day. According to the dossier, "In the year 2021, we had at least 140 (one hundred and forty) murders of transgender people, of which 135 (one hundred and thirty-five) were *travestis* and transsexual women, and 05 (five) cases of transmen and transmasculine people" (ANTRA 2022, p. 30).

6   Reference to the well-known monologue of Agrado in the movie *All about my mother*, by Almodóvar (1999, *Todo sobre mi madre*).

7   Tatton-Schiff reflects on Mollenkott's (2001) idea of "the gender binary construct" and how it grounds "heteropatriarchy".

8   In relation to the difficulties of access to public health care services in Brazil see Rocon et al. (2016). According to the authors the disrespect for the use of the social name, general discrimination and the diagnosis (gender dysphoria) itself are the main obstacles to the access of health care services.

9   According to Vitória Régia da Silva for the website *Gênero e Número* (2021) [Gender and Number], in 2020, the number of medical consultations for gender confirmation surgery process, which is the process of gender transition, fell dramatically. In the Brazilian public health service (SUS) surgeries decreased by 70% and hormone therapy by 6.5% compared to the previous year. The decrease in these procedures in the context of the COVID-19 pandemic makes explicit the social inequalities and how they reflect in the health care provided to trans people—particularly in the context of crisis.

10  Pelúcio (2005) discusses the gender construction of travestis in São Paulo in relation to body modifications. According to her informants, what makes a perfect and most beautiful (*belíssima*) body is being "all broken" (*toda quebrada*).

11  According to Pelúcio (2005, p. 109), *bombadeira* is "a person, usually a travesti, that applies liquid silicone in the body of other travestis or women who seek more bulky or round shapes".

12  The use of the word *tetas* [Portuguese] (tits)—instead of any of its more decent synonyms—points to the ordinary and popular language and reflects a queer/indecent epistemology. It announces that this is an embodied reflection and that it comes from Latin, tropical, erotic bodies, but also bodies that have been regulated, controlled, and domesticated by theologies that still insist on annihilating the experience of the body in the theological *quehacer* (doing). The choice for the word *tetas* has the objective of creating, from the beginning, a queerness—an important rupture that moves the reading from a comfortable place of recognition to a place where the unusual presents itself.

13  "Cystemic" is used in this text as a reference to the "cisgender system" which functions as a normalizing tool for cissexist paradigms.

14  There are many words and expressions in religious studies and theology to talk about that which is experienced in a religious context. "The ultimate concern" (Tillich 1985), "the sacred" or "numinous" (Otto 2007) or the more conventional "transcendent" or "God" are some examples. Even though they are used as analytical concepts (and not to describe a specific or material element), they very easily become symbols (in terms of religion) and are attached to specific images. In this article we use "divine" in the same sense (the matter and expression of religious experience), but also as an adjective to indicate elements where or in which the religious experience can take place.

15  Making explicit the centrality of the body and its relations has been central and an undeniable contribution of Feminist, Gay, Lesbian and Queer Theologies in different contexts. See, for example Ströher et al. (2004).

16  Or as Mary Hunt (2009) states: "bodies don't lie".

17  According to Althaus-Reid (2008, p. 72): "Cosmetic surgery might well claim to stand as a successor to the Christian tradition in its offer to restore a woman's body to its original form, from which ideal, spiritual and aesthetic, it has fallen. Cosmetic surgery

may not be a procedure indicated in the Bible (except perhaps in circumcision) yet the hermeneutical principle is there. Even the most natural functions of the female body are condemned as pollution, requiring ritual acts of purification and restoration".

[18]    On "anti-pedagogies of cruelty" see Segato (2018).

[19]    As response to "alienation and estrangement" (pp. 301–3) the author proposes "liberation and reconciliation" (p. 305) or "flourishing" (quoting Hero 2017).

[20]    In this text Althaus-Reid reflects on a passage of Frida Kahlo's diary, where the painter draws a "fragmented body" and writes about herself *You soy la desintegración* ("I am disintegration"). The article is part of a collection of articles reflecting theologically on the work and live of Frida Kahlo (Eggert 2017).

[21]    Separated by more than 2000 km.

[22]    The ashes refer to the burning down of the building Lorena was at, since she did not burn to death, but because of the smoke she inhaled.

[23]    In a paper presented at the Discernment and Radical Engagement (DARE) Global Forum, Marilu Rojas Salazar (2018) reflected on the common act of burning women's bodies as part of the violence against them in the Mexican context: "Why do they burn the bodies of two autonomous, free and intelligent women? Why do they burn their house, books and vital living surroundings? Is this the new pit fire where to burn the 'witches' that threaten the globalized patriarchal capitalist world? To make disappear the bodies, to violate them or to mutilate them is a way of the patriarchate to punish women" (Rojas Salazar 2018).

[24]    Although it will not be discussed in this article, there are situations in which there is an "authorization" for the public exhibition of breasts in cis-heteropatriarchal society (for example, during carnival festivities). Still, in Brazilian culture even breastfeeding in public is many times questioned. Also, in an analysis of the hagiography and festivities of Saint Agueda of Catania (who was tortured and mutilated in the III century and whose representation and festivities move around her mutilated breasts) Ana Luiza Eiterer Mechler (2021) presents an indecent perspective and states: "The sexual dissidences present in the histories of the virgin martyrs are subjugated and surrounded by a moralism that mutilates and takes away the indecent essence of the real happening or of acts considered holy" (Mechler 2021, p. 51).

[25]    *Tranvestigênere* (Portuguese), according to the Brazilian councilwoman Erica Hilton (Correia 2022), who coined the word, means "all identities of trans men and women, transvestites, non-binary trans people, people who eschew the CIStema".

[26]    See Ivone Gebara's concept of "religious biodiversity" (Gebara 1997) and of an "epistemology of the ordinary life" (Gebara 2015).

[27]    Butler (2009) calls this "equivocality of the statement".

[28]    Marcella Althaus-Reid (2000) reflects extensively on this in the discussion of the idea of decency/indecency in Latin America.

[29]    According to Pereira Torres (2016, p. 92): "By sacralizing the women and profanate the animal, besides the decadence of Christian values and the bridge it established between the West and the East—also expressed in other moments of the song—the inversion of the terms *sacred* and *profane*, could suggest the fall of the aura, as understood by Walter Benjamin (2010), in the sense of the desacralization of the artist—in this case the woman who sings—and his creation. The act of writing gets confused with the insertion in the sacred, but not in the sacred understood religiously, coming from the divine. Here, the sacred became earthly and is materialized in the voice of the woman who sings, sacralized and profanated simultaneously".

[30]    The book *The grace of the world transforms god* (2006) proposed a dialogue with the theme of the 9th Assembly of the World Council of Churches that took place in Brazil in 2006. In response to the theme "God, in your grace, transform the world", the organizers state: "The commas are removed, the words get jumbled up, mishmash. The world is grace and disgrace. Contradiction. Conflict. The world is beautiful and troubled, and grace is to identify the signs of life and live from them, rejecting all sexist, racist, totalitarian and predatory ideologies" (p. 6).

[31]    According to Butler (2009, p. 148): "The disjunction between the statement and the meaning is the condition of possibility to revise the performative, the condition of possibility of the performative as repetition of its first instance, a repetition that is at the same time a reformulation".

[32]    In reference to the expression "love Bethânia" according to Pereira Torres (2016) a reference both to Caetano's sister and Gal's friend Maria Bethânia, the city of Bethany is put in an antagonic pair with Tel Aviv. "Once more, the lyric I distances himself from a more generalized concept of Christendom associated with guilt and redemption, idealizing a love that moves away all suffering represented metonymically by the term *cross*" (pp. 97–98).

[33]    According to Caetano himself: "I was in Europe and, answering a request from Gal, made this song with mutant choruses (sic) that are difficult to memorize. In fact, it is also a song about Gal. Or better: seeks to dialogue with Gal's public persona" (Veloso 2003, p. 74—as cited in Pereira Torres 2016, p. 86).

[34]    In relation to the change of the good/bad milk spilled over the uptights: "Such modification is intensified in the subsequent verses, when nothing from the bad milk is spilled over 'uptights', until finally all, uptights and not-uptights, will receive their dose of good milk, erasing the differences between good and evil, in an absence of value judgment, since all are deserving of the good milk" (Pereira Torres 2016, p. 94).

[35]    Bibliodrama is a way of reading the Bible through different dramatization techniques (See Roese 2007).

[36]    "Chúpatela" is an expression used in the project "Teología Sin Vergüenza" [Theology with no shame], especially at the end of the presentation of each episode/interview produced for the YouTube Channel, by the hosts Lis Valle Ruíz and Alba Onofrio—AKA

*Reverenda Sex*. "[Lis] Here we are in the program Teología Sin Vergüenza—Teología Cuir Feminista which enables justice to flow. How many Christians do you know who live with no shame of being *cuir* [queer] and/or feminist? Most people have never heard of us, but yes we exist and we are here! We are everywhere fighting in the academy, in the churches, in the homes and in the streets. We fight for our communities and for our own lives. We are not machos, but we are *muchas* [many]. [Alba] We are *cuir* feminists with no shame in having faith; we are theologians with no shame of being activists. We are the *sinvergüenzas* ["noshamers"]. We take the juice out of the Bible to produce a *cuir* feminist theology that freshens the fight and enables justice to come. [Lis] In each episode you will find a very theology juicy as papaya water. [Alba] *¡Chúpatela!*" [Suck it up]. More information on the project at https://soulforce.org/teosinverguenza/ (accessed on 27 October 2022).

[37]   (Veloso 1984). Vaca profana, with performer Gal Costa. RCA—103.0637, Vinyl, LP, Album.

[38]   Reference to Belo Horizonte, capital of the State of Minas Gerais, Brazil.

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
