# Peer review of "Divinas tetas: Doing Theology from Mutilated Bodies"

_religions, doi:10.3390/rel14020191_

Round 1
Reviewer 1 Report
Praise
This is a commendable theological extension of Marcella Althaus-Reid’s work on indecent theology. I greatly appreciate the creative and indecent counternarratives from a position of “cutting and pasting,” the consideration of “tetas”, the lyrics of the song “Vacas,” and the focus on trans lives as a locus for theologizing. This is high quality and sound theologizing.
Main Comments
My primary concern, although not a deal-breaker, is that there seems to be a subtle romanticization of trans suffering that interferes with the theologizing. Even though the author uses actual trans experiences for theologizing, the practical dimension and transformative potential of their theology seems to beg to be heard.
Althaus-Reid uses mutilated bodies as a metaphor for indecenting theology, yet she does not actually theologize from mutilated bodies, unlike the present author. I agree that mutilated bodies are subversive imaginations which can inform theology, but what of dashed and unfulfilled dreams, and real human sufferings, injustices and the destruction of lives that are brought about by inaccessibility to transgender support services? How does theology speak to this condition? Is theology only purchased with blood and mangled flesh? Is human life always the cost of queer theologizing? How do liberation theologies benefit those who die brutal deaths, and what do horrible deaths do for the living? It is wonderful to honor and be inspired by the saints and martyrs, but when does dangerous memory end and a fetishization of suffering and death begin?
To some extent, the author may need to challenge Althaus-Reid herself. I invite the author to consider this comment.
Minor Comments
There is an overall need for grammatical and syntactical editing for greater clarity
1. “would we ever need to defend ourselves of someone who actually has such authority?” – this an example of a sentence that lacks clarity.
2. ANTRA’s statement - Is this report in Portuguese or English? If the former, whose translation is it? Please indicate for all instances of translation.
3. “In a decolonial perspective, that is to say the articulation of capitalism, racism and patriarchy (Segato 2021)” – this is an example of a hanging sentence that needs to be rectified for greater clarity.
Author Response
We appreciate the comments and observations and have incorporated those we feel is possible at this time. We have indicated that all translations are our own and worked on grammar issues. The fetishization of suffering is always a risk in relation to liberation theologies, since it’s primary locus for theologizing is the experience of the poor. We would argue, though, that this risk lies primarily in how such theologizing is received, especially in the context of not being familiar with real and concrete suffering. When the experience of suffering is real, it is hard to romanticize it. Thus, the movement from denunciation (of suffering) to annunciation (of justice) in liberation theologies. Vaca profana, in this sense, is a way of announcing a different reality which can be lived out in the lives of trans people already now, but not just yet in its fullness. The struggle continues.
Reviewer 2 Report
There’s a great deal to recommend publication of “Divinas tetas.” First and foremost, it is a clearly written, engagingly presented, well-organized piece of constructive queer/trans theology. Second, it brings into conversation important figures in the field—Marcella Althaus-Reid, Paul Preciado, and Judith Tatton-Schiff; a trio that has something to say to each other, in the author’s hands, but who might not readily be brought together by other writers. Related to this, the author teases out details from Althaus-Reid’s work that can be useful for the ever-growing field of trans theology, making her on-going value for queer/trans theologies all the more apparent. Third, the author engages a tragically compelling story of the death of one trans woman, Lorena Muniz, as well as a probably not-well-known beyond Brazil piece of popular music, “Vaca profana.” Taken together, the author has provided not only a provocative and potentially generative meditation on the need for “divine tits,” but also lifted up a set of materials that might be mined for other constructive theology projects.
There are also ways in which this submission could be strengthened. First, it would be good to add a paragraph somewhere that establishes the representativeness of Lorena Muniz’s experience. Certainly, most people doing queer and trans theologies are fully aware of the precarity of trans women’s lives, but Muniz’s experience seems fairly unique in its specific details and it would be great if she could be shown to actually stand in for a larger problem—about the risks of surgical intervention—rather than just rhetorically stand in for this larger problem. This is particularly notable given that the documentary about the risks of industrial silicone discussed in the submission is from 1987—i.e., 35 years ago, so it’s unclear, without further documentation, that the practice reference even continues to take place, or to what frequency it does. Trans lives are still at risk, but they are certainly not at risk in the same ways.
Second, the treatment of “Vaca profana” would be stronger if it was actually more of a reading of the song. Very little of the song is quoted in the body of the text, although the song’s entire lyrics are helpfully produced and translated in the footnotes. There are some very strong and extended theoretical glosses offered about what this song means--by the author and other commentators—but it’s tough to know how the author gets to those conclusions from the song given what’s written in the text. Or whether the author could get to the same place without the song. If the song is a theological resource, then it needs to be handled better as a resource. Moreover, I wonder about the tension of a cow’s tits and a prosthetic tits—namely, the capacity to produce milk. This capacity is given the most attention in the treatment of the song, but the author notes that this is a specific “limitation” (I should be using a more neutral term here, there’s no intent to malign the “realness” of prosthetic tits by noting their non-lactating character) of prosthetic tits. How do we think this together?
Third, this leads to an omission that should be addressed in revising the submission. There should be some mention somewhere of the visual/textual tradition of the lactating Jesus. Jesus side wound—Jesus’ crucified, mutilated body—is understood as a “divine being” (like a profane cow?) that gives milk (food) to the faithful. Karma Lochrie is one queer-friendly author who has written about this tradition. Given Marcella Althaus-Reid’s interest in folk traditions about Jesus, her theological invocation of a Bi/Christ, and her interest in a desexualization/decorporealization of Mary, it seems like referring to this “classic” theological resource could make the argument stronger.
One other possible addition that could strengthen the essay, although I don’t consider this a necessary tradition in the way I do some extended consideration of the lactating Jesus, is Judith Butler’s notion of the lesbian phallus. At least a footnote relating their consideration of that phallus to Preciado’s discussion of the dildo would be helpful to some readers of this essay.
And maybe one other potential addition. Honduran/Afro-Cuban artist Andres Serrano was known for producing religious imagery that engaged bodily fluids. Most famous for producing the infamous Piss Christ; he also created, during the same period, “lactating” and “bleeding” plexiglass crosses—crosses filled with milk and blood that strategically leaked. He made these pieces (including Piss Christ) in 1987, so just a few years after “Vaca profana.” It could be worth a footnote.
Finally, I would recommend a revision of the opening paragraphs. At the very least a rephrasing, or maybe a reordering of details, and perhaps even a complete omission. Reading an essay that apologizes for/undermines the legitimacy of its very existence on identity grounds at the outset feels kind of beside the point. It will create a resistance/skepticism in some readers from the jump, and the author wants to have the reader on their side. I wonder what kind of essay this would be if the question on page 8, lines 274-75 become the opening lines of the entire piece? (Also, those lines have a word missing about “500 ml”.)
Some minor notes:
Footnote 5: Do we want a fictionalized Spanish trans woman whose words come from a cis gay man filling in the void of speech from a Brazilian trans woman? At the very least, there’s the worry about collapsing all trans people into one entity. At the worst, there’s the potential of making trans women into mute fantasy figures who can be made to speak by us to and for us.
Footnote 12: Love the neologism “cystemic.” I wonder if “cisstemic” is a better spelling?
Footnote 16: This quote from Althaus-Reid is really powerful. I wonder if it could be moved to the text? But then there’s also the worry of presenting gender-affirming surgery as cosmetic surgery.
Page 6, line 199: “Tattoh-Schiff” should be “Tatton-Schiff”
Page 6, line 210-11. This sentence is unclear. It seems to be intended as a recapitulation of the prior sentence, but I had to read a few times to get that. And if it's not intended as a recapitulation, then I don't know what it means--it's incomplete.
Page 6, line 218. Here and one time later, the language “dreamed of” is used. I’m not troubled by this language, but I wonder about the implication of Muniz (trans people generally) being invested in a fantasy of gender. I mean, technically, we are all only ever invested in a fantasy of gender, but spelling out what that means for trans politics is worthy of its own article (if not several articles), so the wording might need massaging. I also wonder about the implications of “fragmentation” in the Althaus-Reid quote. Again, I think the more we emphasize a universal fragmentation that longs for coherence the better off we are, but given that “fragmentation” has an evaluative tinge, maybe think about the uncareful reader, especially in the fraught terrain the essay covers.
Page 7, line 224. I think the point of the final sentence of the section is more powerful if there is a space between “every” and “body.”
Page 7: I found the example of Indianare Siqueria as interesting if not more powerful theologically and politically than the example of Muniz. I wonder if there’s a way to develop this example—fill it out a bit, was there any press coverage/commentary? Or move it, present it in some way that it doesn’t get lost in the texture/flow.
Page 8, line 296: I love slash and parenthetical constructions, but they have to be done well. It should be “il/logic.”
Page 9, line 332: Should there be a “the” before “year” in the quotation about the composition/publication details of “Vaca profana”? And do we need all these details about the composition/publication? It seems out of balance to the absence of close reading of the song itself.
Page 11, line 383: “heard” should be “herd”
Author Response
We appreciate the careful and engaged reading expressed through the comments and suggestions. We tried to incorporate as much as possible and whenever we saw it appropriate.
We have tried to make it more clear that the experience of Lorena is very common in the lives of trans people in Brazil adding some sentences in the text. Also, there are other more recent references mentioned in the article that discuss both the general situation of trans people in Brazil and specifically the issues of surgeries and use of industrial silicone.
We have tried to let both Lorena’s story and the lyrics of Vaca profana speak from themselves and present how they motivate our theological doing. There is no intention in “milking” all the possible ways in which they can be read or be engaged. The fact that it relates to the Brazilian context more immediately might not make this evident in other contexts, but it is hard to exhaust all the contextual subtleties. We do believe there is a general flow and reflection that enables the understanding of how we think of them and how they inform our own construction. We are not sure how the journal deals with copyright issues. That is why we opted for not quoting long parts of the song in the text and presented the full lyrics at the end as an epilogue.
We appreciate the indication of the references and will look at them for future reflections. But we understand that, since we are not familiar with them and they would open other windows for reflection, it is better to leave them out at this time.
We don’t see the issue of making clear that we do not have the experience of having breast implants as apologetic or undermining the legitimacy, but of intellectual and political honesty. We think it is fundamental that people know from the start that we are reflecting on the issues raised in relation to how they affect us. This is especially important for us considering that we are very close, work with and respect deeply trans people.